# Deciphering the Role of microRNAs: Unveiling Clinical Biomarkers and Therapeutic Avenues in Atrial Fibrillation and Associated Stroke—A Systematic Review

**DOI:** 10.3390/ijms25105568

**Published:** 2024-05-20

**Authors:** Elke Boxhammer, Christiane Dienhart, Richard Rezar, Uta C. Hoppe, Michael Lichtenauer

**Affiliations:** 1Department of Internal Medicine II, Division of Cardiology, Paracelsus Medical University of Salzburg, 5020 Salzburg, Austriau.hoppe@salk.at (U.C.H.); m.lichtenauer@salk.at (M.L.); 2Department of Internal Medicine I, Division of Gastroenterology, Hepathology, Nephrology, Metabolism and Diabetology, Paracelsus Medical University of Salzburg, 5020 Salzburg, Austria

**Keywords:** atrial fibrillation, biomarker, cerebrovascular disease, microRNA, ischemic stroke

## Abstract

MicroRNAs (miRNAs) are small non-coding RNA molecules that regulate gene expression by binding to target messenger RNAs (mRNAs). miRNAs have been implicated in a variety of cardiovascular and neurological diseases, such as myocardial infarction, cardiomyopathies of various geneses, rhythmological diseases, neurodegenerative illnesses and strokes. Numerous studies have focused on the expression of miRNA patterns with respect to atrial fibrillation (AF) or acute ischemic stroke (AIS) However, only a few studies have addressed the expression pattern of miRNAs in patients with AF and AIS in order to provide not only preventive information but also to identify therapeutic potentials. Therefore, the aim of this review is to summarize 18 existing manuscripts that have dealt with this combined topic of AF and associated AIS in detail and to shed light on the most frequently mentioned miRNAs-1, -19, -21, -145 and -146 with regard to their molecular mechanisms and targets on both the heart and the brain. From this, possible diagnostic and therapeutic consequences for the future could be derived.

## 1. Introduction

### 1.1. Prevalence of Atrial Fibrillation and Ischemic Stroke

Atrial fibrillation (AF) is a predominant cause of cardiovascular diseases, and acute ischemic stroke (AIS) are among the most common cerebrovascular diseases, both bearing significant prevalence on global scale [1]. According to estimates from the World Health Organization, cardiovascular diseases account for 17.9 million deaths each year, with AF and its consequential cardiovascular and cerebrovascular ramifications being the leading causes of death, disability and reduced quality of life [2].

AF is associated with a wide range of risk factors, including hypertension, diabetes, obesity, smoking and alcohol consumption [3]. The prevalence of AF and AIS is expected to increase, as the population ages and more people are exposed to risk factors [4]. In addition, AF is also associated with other comorbidities, such as congestive heart failure [5] and valvular heart disease [6], which further increase the risk of AIS in these patients.

Mortality and morbidity rates of AF and associated AIS are particularly prominent in low- and middle-income countries, where access to diagnosis and treatment is limited. As such, identifying and managing risk factors is essential to reducing the burden of AF and stroke in these countries. There is also a need to improve public awareness and education about the signs and symptoms of AF and stroke, to encourage early diagnosis and treatment [7].

### 1.2. Pathophysiology and Prevention

AF is an arrhythmia caused by disorganized electrical activity of the atrium, leading to an irregular and rapid heart rate. This can cause blood to pool and stagnate in the atrium, increasing the risk of clot formation. If a clot breaks off and is carried through the bloodstream, it can cause an AIS. A stroke occurs when a blood vessel in the brain becomes occluded and lack of oxygen and nutrient-rich blood induces brain cell death [8,9].

The precise pathophysiological mechanisms of AF and concomitant AIS have yet not been fully understood, as previous studies could not demonstrate a definitive temporal correlation in the concurrent manifestation of both AF and AIS [10]. However, several risk factors have been identified that can increase the likelihood of both AF and stroke. These include increasing age, hypertension, hyperlipidemia, diabetes and sleep apnea [3]. An increased risk can also be associated with gender, family history and lifestyle factors such as smoking, alcohol use and a sedentary lifestyle [11].

Research has found that AF can lead to an increased risk of stroke due to multiple mechanisms. These include an elevated risk of thrombus formation and embolism, reduced cardiac output and impaired left ventricular diastolic filling [12]. Additionally, AF can lead to an increased risk of atrial dilatation, which can further increase the risk of stroke.

In order to reduce the risk of stroke in patients with AF, lifestyle modifications such as reducing alcohol intake, quitting smoking and increasing physical activity are important. Additionally, a number of medications have been found to reduce the risk of stroke in patients with AF. These include anticoagulants like non-vitamin K oral anticoagulants (apixaban, dabigatran, edoxaban, rivaroxaban) and warfarin [13].

### 1.3. miRNAs in Atrial Fibrillation and AIS

MicroRNAs (miRNAs) are a class of small regulatory RNA molecules, approximately 22 nucleotides in length, that are involved in post-transcriptional gene regulation [14,15]. They are known to regulate the expression of other genes in a variety of ways, including by blocking the translation of certain messenger RNAs into proteins or by degrading them. miRNAs have been found to play a role in many physiological processes, such as development, stress response, regulation of immune system and circadian rhythm, metabolism and aging [16,17]. miRNAs are encoded by the genome in the form of primary transcripts, which are then further processed by the RNA interference (RNAi) machinery to form mature miRNAs. These mature miRNAs are then incorporated into the Argonaute family of proteins and loaded into the RISC complex [18], where they can bind to their target mRNAs and regulate their translation.

The field of research on miRNAs has expanded considerably in recent years, as they are believed to be involved in the development of many diseases, including cardiovascular disease. Recent studies have suggested that miRNAs may play a role in the development and progression of AF [19]. The role of miRNAs in AF is complex and involves multiple pathways. miRNAs were detected to regulate cardiac structural remodeling, ion channel function and Ca^2+^ circulation, as well as inflammation and fibrosis [20,21], all of which are involved in the development of AF. For example, miRNA-21 is upregulated in the myocardium of patients with AF in contrast to patients without AF, and it has been shown to promote the development of AF by modulating gap junction connexin 43 and cytoskeletal proteins [22]. Further, miRNA-208 has been found to be upregulated in the atria of AF patients and promotes AF development by targeting SERCA2a, a key regulator of the Ca^2+^ cycle [23]. In addition, miRNAs play an important role in the inflammation and fibrosis associated with AF. For example, miRNA-133a was downregulated in several animal models in the atria of AF patients and supports fibrosis by targeting TGF-β1 and TGF-β receptor expression [24].

Similarly, in the context of AIS, miRNAs seem to play a key role due to their involvement in cellular and molecular processes (inflammation, oxidative stress, apoptosis). miRNA-21 is increased in the brain after AIS and regulates the expression of proapoptotic and inflammatory genes such as Bcl-2-associated X protein (Bax) and TNFα [25]. miRNA-146a is upregulated in response to ischemic injury and can regulate the expression of inflammatory and anti-apoptotic genes such as NF-κB and Bcl-2 [26]. In addition, miRNAs are associated with the regulation of autophagy, a cellular process important for maintaining cellular homeostasis and preventing cell death [27]. In AIS, autophagy is often disrupted, so several miRNAs, such as miRNA-133a and miRNA-244-3p, regulate autophagy-related genes and signaling pathways [28]. Furthermore, miRNAs are also associated with the regulation of angiogenesis, which is important for restoring blood flow to the brain and is often impaired in AIS. Several miRNAs, such as miRNA-210 and miRNA-155, are involved in the regulation of angiogenesis-related signaling pathways and genes, such as VEGF and eNOS [29].

With regard to the etiology of elevated miRNA concentration, a prevailing pathomechanistic paradigm that holds fundamental significance in AF and AIS revolves around the phenomenon of hypoxia. This notion was initially substantiated by the discernible upregulation of hypoxia-inducible factor (HIF) and subsequent HIF-mediated modulation of miRNA expression. Subsequent investigations have further demonstrated that hypoxia exerts a direct regulatory influence on miRNA processing, that may transpire through pathways independent of HIF [30,31].

Moreover, AF and AIS are shown to overlap in predisposing factors, including but not limited to hypertension, diabetes mellitus and advanced age, which might have the potential to induce secondary impairments encompassing endothelial dysfunction, heightened oxidative stress and inflammatory cascades. These cumulative perturbations may culminate in an increase of miRNA expression. In light of this, it is conceivable that the shared risk factors might contribute to the observed parallel elevation of miRNA levels in both AF and AIS [32,33].

Taken together, miRNAs play a critical role in the pathogenesis of both AF and AIS by regulating multiple cellular and molecular signaling pathways. Understanding the role of miRNAs in both often related pathologies may help identify potential therapeutic targets for treatment.

### 1.4. Aim of the Review

While numerous studies have extensively examined the relationship between AF and miRNA [19,34,35,36,37,38], as well as cerebrovascular ischemia/stroke and miRNA [39,40,41,42,43], there remains a conspicuous gap in the literature regarding the combination of AF, cerebrovascular events and miRNA. Indeed, only a scant number of publications have delved into this intersection. Consequently, the primary objective of this review is to consolidate existing knowledge on miRNAs, emphasizing their potential role not only as pertinent biomarkers for the early detection of cardioembolic events but also as promising therapeutic targets following AIS associated with AF.

## 2. Methods

A systematic database research was performed in PubMed Central^®^ from January 2024 to March 2024 with only English-language articles (original work and suitable reviews) included in this overview. The following keywords were used in combination: miRNA AND atrial fibrillation AND stroke/cerebrovascular disease or miRNA AND atrial fibrillation-related stroke/cardioembolic stroke. To filter out appropriate studies for this review, corresponding abstract were screened in addition to the title. Publications included were read in their entirety, whereas duplicate manuscripts were excluded. Reference lists of considered studies were also checked for further readings. This review was conducted based on the Preferred Reporting Items for Systematic Reviews and Meta-Analyses (PRISMA) guidelines [44] (Figure 1).

## 3. Results

### 3.1. Overview of All Described miRNAs

Of 103 papers identified using database research, 19 studies were ultimately included in the review presented here. These contained one in vivo study, one combined human and in vitro study, 4 reviews and 13 isolated human studies. A tabular overview of the included papers with the respective authors, year of publication, miRNAs involved, type of study and major outcomes can be found in Table 1.

As it can be seen from this table, there are a large number of miRNAs that have been studied in the context of AF and acute embolic/ischemic stroke. However, a large proportion of the miRNAs listed here have been described in only one study and do not find resonance in other studies, thus lacking appropriate comparison. In Zou et al. [61] or Xie et al. [59], for example, miRNAs-27a, -27b, -30e and -494 were investigated, but these are not repeated in any other of the papers thematically included here.

In addition, many studies did not use miRNAs as preventive markers for the possible avoidance of AIS as a consequence of AF. Rather, plasma levels were measured after AF and descriptive statements were given about the expression level of the respective miRNAs [47,48,55]. From this point of view, it is very difficult to derive a therapeutic–medicinal consequence. In contrast, the group of Sadik et al. [57] could show that the upregulation of miRNA-155 had a high diagnostic accuracy for the occurrence of AIS in AF. Also, Kiyosawa et al. [49] were able to place numerous miRNAs in positive or negative correlation with the CHA_2_DS_2_-VASc-Score, optimizing the prediction of potential stroke. These statements are important for the clinical routine, because patients with such a constellation need close-meshed care and optimized anticoagulation.

In summary of the present review, miRNA-1, -19, -21, -145 and -146 were most frequently mentioned in the existing 19 studies; therefore, these miRNAs were subsequently analyzed in more detail with regard to their molecular structures both in the context of AF in the heart and in the context of AIS in the brain to finally weigh therapeutic potentials.

### 3.2. miRNA-1

#### 3.2.1. miRNA-1 Heart

Recent studies showed that miRNA-1 was significantly upregulated in the atria of AF patients. Furthermore, it could be demonstrated that the increased expression of miRNA-1 was associated with an increased risk for AF [50] and AIS [51]. These findings suggest that miR-1 may be involved in the development of AF, at least in part, through its effects on the expression of its target genes (Figure 2).

Research has shown that miRNA-1 and one of its target gene GJA1 plays a role in the development of AF. The GJA1 gene is located on chromosome 6 and encodes for the protein connexin 43. This protein is a gap junction, which allows for direct intercellular communication between cardiac myocytes. This gap junction is involved in the regulation of the electrical properties of the heart, including the conduction of action potentials [62]. The expression of connexin 43 is decreased in the atria of patients with AF, and the decreased expression of connexin 43 has been linked to increased risk of AF [63]. Studies have shown that an upregulation of miRNA-1 during AF is able to decrease expression of GJA1 and consequently connexin 43. The mechanism by which miRNA-1 may act on GJA1 is not completely understood, but research suggests that it is related to alterations in cAMP and IP3 pathways, as connexin 43 is an important structural protein for the exchange of relevant messengers such as cAMP, IP3 and Ca^2+^ between neighboring cardiomyocytes.

Another target of miRNA-1 is the potassium channel gene KCNJ2, also known as the G-protein inward rectifying potassium channel 3 (GIRK3) [62]. It is involved in the regulation of action potential repolarization and has been implicated in the development of AF. KCNJ2 located on chromosome 17 encodes for Kir2.1, which is an inward rectifying potassium channel 1 (IRK1) and a member of the Kir family of potassium channels [64]. Its main purpose is to maintain and regulate membrane potential. Upregulation of miRNA-1 in the context of AF leads to suppression of KCNJ2 and subsequently to decreased expression of Kir2.1 channels. The silencing of these potassium channels are associated with development of AF, as the decreased activity of these channels leads to an increased risk for arrhythmias.

MiRNA-1 additionally affects several genes involved in the regulation of cardiac ion channels and thus electrical activity, including the transcription factor IRX5 and the sodium channel gene SCN5A. The transcription factor IRX5 is downregulated in atrial tissue samples from patients with AF [65]. Moreover, IRX5 is a respective regulator of ion channel expression, including the voltage-gated sodium channel Na_V_1.5, and its loss increases the risk of AF. Additionally, miRNA-1 directly targets SCN5A, a regulator of the electrical properties of cardiac cells. Studies have demonstrated that the downregulation of SCN5A caused by miRNA-1 causes an increase in Na_V_1.5 expression, which in turn contributes to the pathogenesis of AF [66]. Finally, Na_V_1.5 plays an important role in the electrical activity of the heart. Studies have shown that increased expression of Na_V_1.5 is associated with a higher risk of AF. Thus, it should be of great interest to further elucidate to what extent miRNA-1 is involved in the pathogenesis of AF through its presumed regulation of IRX5, SCN5A and Na_V_1.5 and whether this modulation is perhaps causally linked to AF. However current suggestions and evidence indicate that a drug-induced downregulation of miRNA-1 would likely be associated with anti-arrhythmic effect and lower risk of subsequent cardiovascular events [46].

#### 3.2.2. miRNA-1 Brain

There are few data in the current literature on the molecular mechanisms of miRNA-1 during and after AIS, which have been obtained mainly from animal models. Using a recent study [67] in a human collective, it was shown that plasma levels of miRNA-1-3p were associated with subclinical atrial fibrillation in patients with cryptogenic stroke, perhaps involving persistent cardiac hypoxia contributing to the observed increase. In an animal model of induced AIS by Jiang et al. [68], it was demonstrated not only that miRNA-1 was significantly upregulated but also that the extracellular signal-regulated kinase 1 and 2 (ERK1/ERK2) pathway was simultaneously downregulated. At the nucleus level, genes for anti-apoptotic activity are decreased when encoded by the reduced kinase activity of ERK1 and ERK2. On the other hand, this animal experiment additionally demonstrated an increase in Bax as a pro-apoptotic protein as well as several caspases (especially caspase 3), which were even more enhanced when expressed by additional administration of an agomir of miRNA-1 [68]. Thus, it can be postulated that miRNA-1 not only upregulates pro-apoptotic factors in ischemic cerebral infarction but also simultaneously downregulates the expression of anti-apoptotic ones, thus, on balance, causing neuroptosis (Figure 3).

### 3.3. miRNA-19

#### 3.3.1. miRNA-19 Heart

MiRNA-19a/b, which is upregulated in atrial fibrillation, appears somewhat contradictory in its effect on the heart (Figure 4). On the one hand, similar to miRNA-1, there is decreased expression of connexin 43 and thus possibly decreased exchange of important transport proteins and relevant secondary messengers such as cAMP and calcium, leading to a possible pro-arrhythmogenic effect [69]. On the other hand, miRNA-19 also inhibits the phosphatase and tensin homolog (PTEN), as well as miRNA-21, which will be described later. Among these, there is increased activation of phosphoinositide 3-kinases (PI3K) and thus protein kinase B, better known as Akt, which in turn inhibits anti-apoptotic processes that occur via Bax or via a corresponding caspase cascade [70]. Therefore, miRNA-19a/b is not only pro-arrhythmogenic and thus in principle “nourishing” with respect to AF, but in parallel makes cardiomyocytes more stress resistant with regard to the underlying rhythmological disease through anti-apoptotic processes.

#### 3.3.2. miRNA-19 Brain

Upregulation of miRNA-19a in AIS [58] triggers downregulation of genes encoding adiponectin receptor 2 (AdipoR2). AdipoR2 is a G-protein-coupled receptor involved in the regulation of glucose and fatty acid metabolism and inflammation. Downregulation of AdipoR2 by miRNA-19a may lead to an increase in oxidative stress and inflammation, which is detrimental to the brain after AIS [71]. The onward cascade proceeds through peroxisome proliferator-activated receptor alpha (PPARα), which is also reduced in activity by decreased AdipoR2 level. Inactivation of PPARα results in increased transmembrane incorporation of intercellular adhesion molecule 1 (ICAM1) and vascular cell adhesion molecule 1 (VCAM1), both of which are important in regulating cell adhesion, particularly leukocyte recruitment and subsequent chemokine release [72]. The upregulation of these genes and proteins by miRNA-19a initiates oxidative stress and neuroinflammation with consequent increased cell death due to ischemia (Figure 5).

### 3.4. miRNA-21

#### 3.4.1. miRNA-21 Heart

MiRNA-21 is upregulated in the atria of patients with AF and is associated with a downregulation of relevant genes regulating Ca^2+^ homeostasis [22], especially the calcium voltage-gated channel subunit alpha1 (CACNA1) and the calcium voltage-gated channel auxiliary subunit beta (CACNB2). These genes encode for two relevant calcium channel subunits, which are both integral membrane proteins and form long-lasting calcium channels (CaV1.2). The consecutive downregulation of these two proteins leads to a decrease in the number and activity of CaV1.2 channels, which has been associated again with decreased cardiac excitability and decreased calcium influx, resulting in impaired contractile force and arrhythmias [73].

MiRNA-21 is also linked to the regulation of the sprouty1 (SPRY1) gene. SPRY1 is a negative regulator of the extracellular signal-regulated kinase (ERK)-mitogen-activated protein kinase (MAPK) pathway, which is a major signaling cascade and involved in the regulation of cellular processes such as proliferation, differentiation and apoptosis. In the case of AF, this signaling cascade pathway is essentially involved in development of fibrosis and collagen formation (collagen 1 and 3). Increased collagen production, particularly in the left atrium, was associated with a significantly increased risk of AIS in a paper by Marrouche et al. [74]. The increased fibrosis of the heart, which is ultimately associated with the progression of AF to atrial cardiomyopathy, is regulated by another molecular mechanism, the CADM1/STAT3 pathway. In the case of CADM1, miRNA-21 has been shown to interact with its mRNA, resulting in the downregulation of its expression. This, in turn, leads to the activation of STAT3 and the subsequent activation of various downstream targets, including those involved in fibrosis and collagen production. This mechanism was further investigated by the working group of Cao et al. [75] with regard to cardiac fibrosis formation, where cardiac fibroblasts transfected with an antagomir of miRNA-21 ultimately showed significantly increased proliferation and collagen production (Figure 6).

#### 3.4.2. miRNA-21 Brain

miRNA-21 is upregulated in patients with AIS [51,60] in laboratory chemical analyses and appears to have an anti-apopteutic and thus neuroprotective effect [52]. In this regard, multiple molecular pathways are involved. The upregulation of miRNA-21 in the context of hypoxia inhibits the programmed cell death 4 (PDCD4) gene and subsequently reduces the expression of the transcription factor Activator protein 1 (AP-1) and the helicase eukaryotic initiation factor-4A (elF4A) [76,77]. At the nuclear level, this constellation leads to decreased expression of pro-apoptotic genes and thus to reduced neuroptosis.

Further, miRNA-21 is a crucial component in the molecular pathway of PTEN, PI3K, protein kinase B also known as Akt and caspase 9, playing in their entirety an active role in neuroprotection after AIS. miRNA-21 primarily reduces the activity of PTEN as a direct antagonist of PI3K. By inhibiting the phosphatase PTEN, the membrane lipid phosphatidylinositol-3,4,5-triphosphate (PIP3) is no longer further degraded to phosphatidylinosital-4,5-biphophate (PIP2) [78]. The amount of PIP3 is even further increased by phosphorylation of PIP2 by PI3K, which in turn activates the serine/threonine kinase Akt. Akt in turn inhibits pro-apoptotic protein such as Bax and caspase 9 [25].

Finally, the reflective increase of miRNA-21 in the context of stroke additionally causes a decreased expression of the Fas receptor, which is a type-I transmembrane protein that plays an important role in programmed cell death. It is also known as CD95 or APO-1, and is expressed on the surface of cells in a variety of tissues, including neurons. By reducing the transmembrane density of Fas receptors, the activity of the pro-apoptotic caspase cascade (caspase 8 → 3 → 9) and the pro-apoptotic protein Bax is significantly reduced, thus preventing increased death of astrocytes after infarction [77] (Figure 7).

### 3.5. miRNA-145

#### 3.5.1. miRNA-145 Heart

miRNA-145 has been identified as a key regulator of AF. The influence of miRNA-145 on AF is mediated through its effects on several pathways, such as the calcium/calmodulin-dependent protein kinase II (CaMKII), the sarcoplasmic reticulum calcium ATPase (SERCA), ryanodine receptors (RyR) and inward calcium channels. The CaMKII pathway is one of the major pathways affected by miRNA-145 and is the central player in AF, as it is involved in the regulation of excitation-contraction coupling and calcium signaling [79,80] (Figure 8).

One of the reasons for the development of AF is a considerable variation of the calcium content (Ca^2+^) both in the cytoplasm and in the sarcoplasmic reticulum (SR), which is explained by the reduced uptake of Ca^2+^ into the SR by SERCA on the one hand and by an increased release via stimulation of the RyR on the other hand. miRNA-145 is downregulated in the context of AF and thus loses its inhibitory influence on the activity of CAMKII [81]. This results in increased phosphorylation of RyR with associated increased diastolic SR-Ca^2+^-release (“leak”). This process is enhanced by additional phosphorylation of inwardly rectifying calcium channels (including CaV1.2) by CaMKII and a concomitant additional influx of calcium into the cytosol. This excessive presence of calcium in the cytosol, particularly in diastole, results in arrhythmia-inducing delayed afterdepolarization by overloading the Na^+^/Ca^2+^ exchanger protein (NCX). In addition, CaMKII appears to affect calcium reuptake into the SR via SERCA. Whether this process occurs molecularly in such a way that the CaMKII directly activates the SERCA or primarily phosphorilates the inhibitory component, namely phospholamban (PLN), and thus the SERCA is secondarily released, is currently the state of research. In both cases, an increased SERCA activity occurs [82,83], which accordingly attempts to reabsorb the increased Ca^2+^ into the SR. These Ca^2+^ fluctuations result in significant disturbances in the excitation-contraction coupling, which ultimately feed the AF and cause cardiomyopathies.

#### 3.5.2. miRNA-145 Brain

miRNA-145 regulates the expression of proteins involved in AIS, including CaMKII and aquaporin 4 (AQP4). Studies have suggested that miR-145 is upregulated in response to AIS [53,55] and acts to downregulate CaMKII expression, which is involved in neuronal plasticity. The reduction of CaMKII activity leads to a relevant reduction of aquaporins, in this particular aquaporin 4 (AQP4), which are transmembranally deposited on the astrocytes. AQP4 is responsible for the transport of water across cell membranes and causes cell swelling and edema formation in AIS. A lower expression of AQP4 leads to a reduced water influx into the cell and thus to a reduced edema formation and probably also to an improved neurological outcome [50,54] (Figure 9).

### 3.6. miRNA-146a Heart and Brain

miRNA-146a is a microRNA that is downregulated in AF. This downregulation has been linked to the activation of the IRAK1/TRAF6 and NF-κB pathways, which are involved in inflammation and cytokine release. The IRAK1/TRAF6 pathway is a signaling cascade that is activated by the toll-like receptor (TLR) family of proteins [84]. When activated, this pathway leads to the activation of NF-κB, which is a transcription factor that regulates the expression of genes involved in inflammation and immune responses. miRNA-146a as negative regulator for inflammation has been found to be downregulated in AF, which suggests that this pathway is activated in AF. The activation of the IRAK1/TRAF6 and NF-κB pathways leads to the release of pro-inflammatory cytokines, such as interleukin-6 (IL-6), interleukin-1β (IL-1β) and tumor necrosis factor-alpha (TNF-α). These cytokines can lead to further damage to the heart tissue and contributes to progression of AF. Additionally, inflammation triggers coagulation, decreases the activity of natural anticoagulants and impairs the fibrinolytic mechanisms. Inflammatory cytokines are the main mediators implicated in the activation of coagulation. Therefore, it is not surprising that a decreased level of miRNA-146a and an associated activation of the aforementioned inflammatory cascade increases the risk of major cardiovascular events (MACE) [45,56,85], especially AIS. The molecular pathway of miRNA-146a in AIS in the brain responds via the same kinases or transcription factors as in the heart. Through the increased expression of proinflammatory genes or second messengers and ultimately the invasion of proinflammatory cells results in post-ischemic neuroinflammation (Figure 10).

## 4. Outlook and Conclusions

The research field of miRNAs has experienced rapid growth in recent years, not only in the diagnostic area, but also beginning to be used therapeutically. Nevertheless, the work on miRNAs with regard to AF and AIS in terms of prevention or even therapeutic potential is very sparse. In this review, not only the thematic studies could be included, but the most frequently mentioned and investigated miRNAs could be highlighted in more detail regarding their molecular mode of action on heart and brain in the context of AF and AIS.

### 4.1. Outlook

Considering the molecular pathways highlighted in this manuscript, the following therapeutic strategies can be hypothesized:Targeting miRNA-1 with an antagomir, which is a chemically engineered oligonucleotide designed to silence specific miRNAs, may stabilize cardiac membrane potentials and rhythmological conditions while reducing apoptotic mechanisms in the brain, thus preventing neuroptosis.Lowering the plasma level of miRNA-19 by antagomir would decrease oxidative stress and neuroinflammation in the brain and stabilize the rhythmological situation in the heart, but make cardiomyocytes more susceptible to controlled cell death.Similarly, the use of an agomir of miRNA-146, which is a synthetic miRNA mimic that enhances the activity of specific miRNAs, can reduce induced inflammation/coagulation in the heart and neuroinflammation in the brain simultaneously.An agomir of miRNA-145, for example, could further enhance the beneficial effect in the brain with reduced edema formation due to reduced transmembrane incorporation of aquaporin 4, whereas in the heart it induces more balanced calcium homeostasis with improved rhythmic control.Only miRNA-21 acts oppositely on the heart and brain with respect to drug therapy regimens, as lowering miRNA-21 levels would prevent increased collagen formation and thus fibrosis of the cardiac situation, but in the brain it would increase the apoptotic tendency of astrocytes in the event of an AIS.

### 4.2. Potential Research Scenarios

Future research in miRNAs should focus on several key directions:Validation and Mechanistic Studies: Conducting extensive in vitro and in vivo studies to validate the molecular mechanisms and therapeutic potential of specific miRNAs in AF and AIS. This involves creating animal models to observe the effects of miRNA modulation on cardiac and brain tissues, focusing on pathways related to inflammation, apoptosis and cellular homeostasis.Drug Development and Delivery: Developing miRNA-based drugs such as antagomirs and agomirs and refining their delivery systems to ensure targeted and efficient treatment with minimal off-target effects. This includes exploring nanoparticle-based delivery methods and assessing the pharmacokinetics and pharmacodynamics of these miRNA therapeutics.Clinical Trials: Initiating phase I and II clinical trials to evaluate the safety, dosage and preliminary efficacy of miRNA-based therapies in humans. These trials would provide critical data on how these therapies perform in real-world scenarios and help refine therapeutic strategies.Interdisciplinary Research: Fostering collaboration between molecular biologists, cardiologists, neurologists and pharmacologists to integrate findings from basic research with clinical insights. This interdisciplinary approach is crucial for developing comprehensive treatment protocols and ensuring that laboratory discoveries translate effectively into clinical practice.Personalized Medicine: Investigating the role of miRNAs in personalized medicine by identifying miRNA profiles that predict patient responses to treatment. This could lead to the development of personalized therapeutic regimens that optimize efficacy and minimize adverse effects based on an individual’s miRNA expression patterns.

### 4.3. Conclusions

While miRNAs hold significant promise for the treatment of AF and AIS, substantial research is needed to transform these speculative therapeutic strategies into clinically viable treatments. The potential for miRNA-based therapies to revolutionize the management of cardiovascular and neurological diseases underscores the importance of continued investment and investigation in this burgeoning field. By pursuing the outlined research directions, we can better understand the therapeutic potential of miRNAs and develop effective, targeted treatments for these conditions.

## Figures and Tables

**Figure 1 ijms-25-05568-f001:**
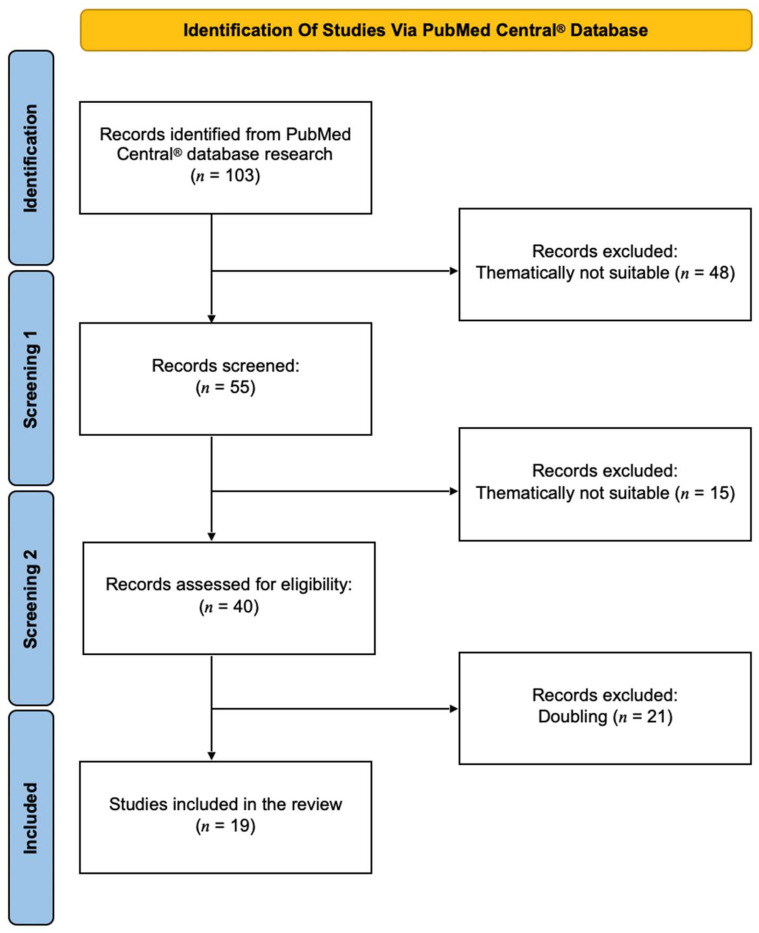
Flow diagram of the database search, screening and inclusion of the studies (modified from the Preferred Reporting Items for Systematic Reviews and Meta-Analyses (PRISMA) guidelines).

**Figure 2 ijms-25-05568-f002:**
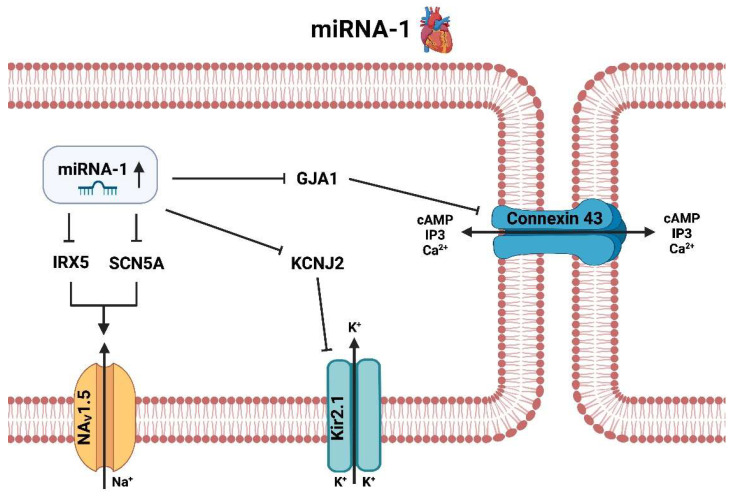
Molecular pathways of miRNA-1 in the context of atrial fibrillation in the heart (created with BioRender.com).

**Figure 3 ijms-25-05568-f003:**
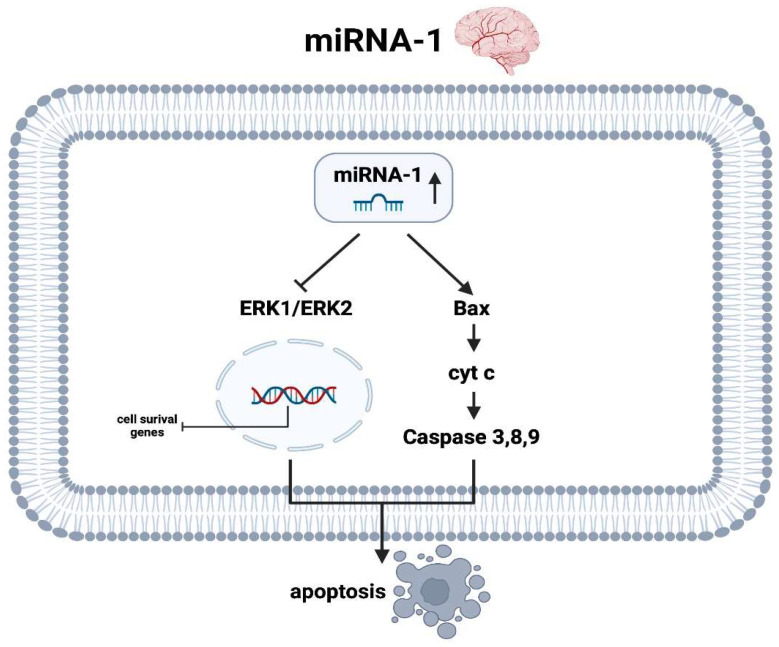
Molecular pathways of miRNA-1 in the context of ischemic stroke in the brain (created with BioRender.com).

**Figure 4 ijms-25-05568-f004:**
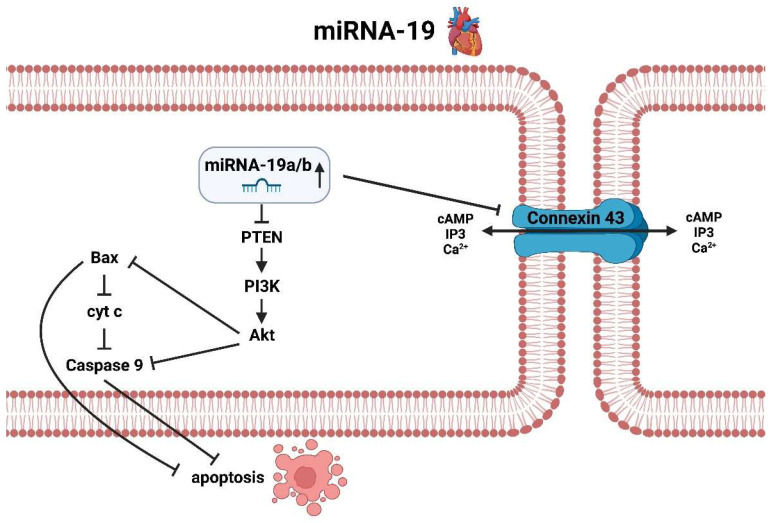
Molecular pathways of miRNA-19 in the context of atrial fibrillation in the heart (created with BioRender.com).

**Figure 5 ijms-25-05568-f005:**
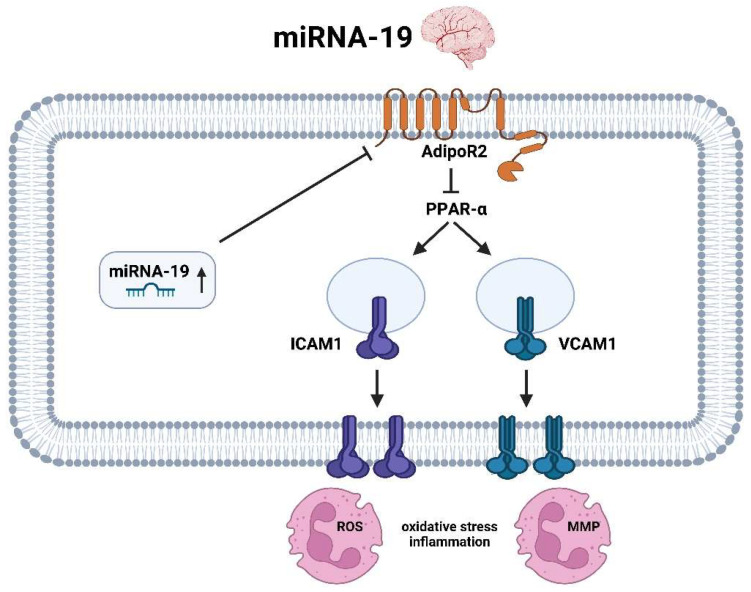
Molecular pathways of miRNA-19 in the context of ischemic stroke in the brain (created with BioRender.com).

**Figure 6 ijms-25-05568-f006:**
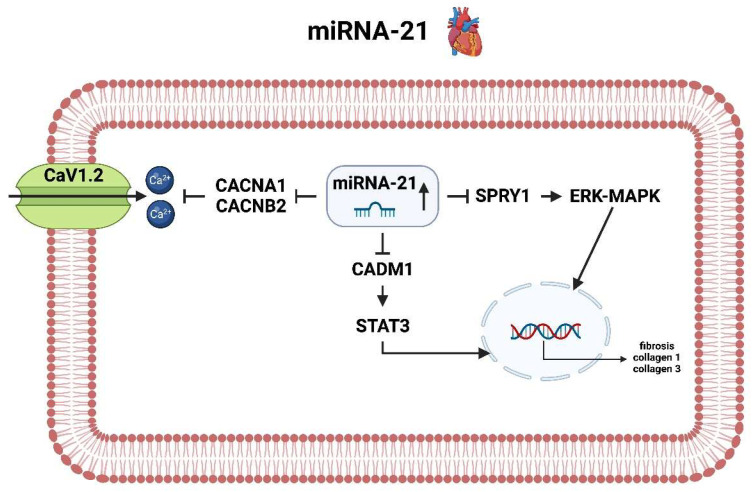
Molecular pathways of miRNA-21 in the context of atrial fibrillation in the heart (created with BioRender.com).

**Figure 7 ijms-25-05568-f007:**
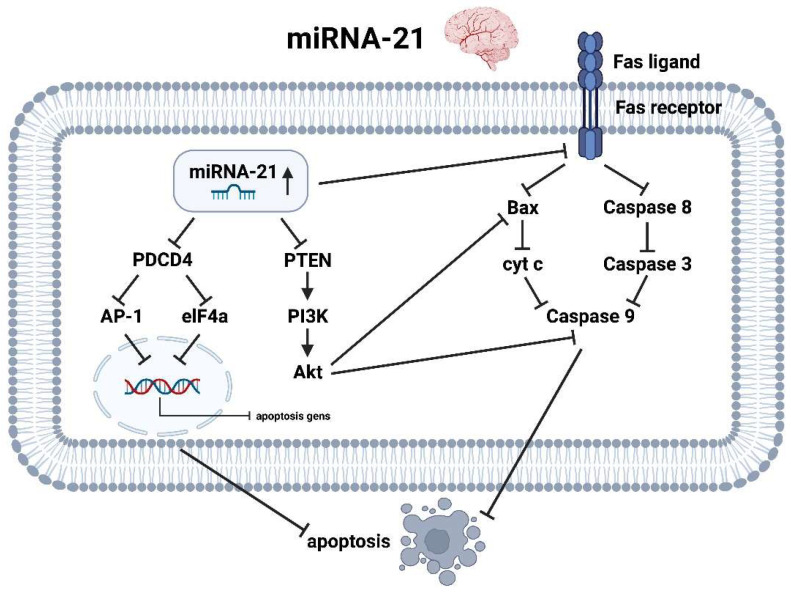
Molecular pathways of miRNA-21 in the context of ischemic stroke in the brain (created with BioRender.com).

**Figure 8 ijms-25-05568-f008:**
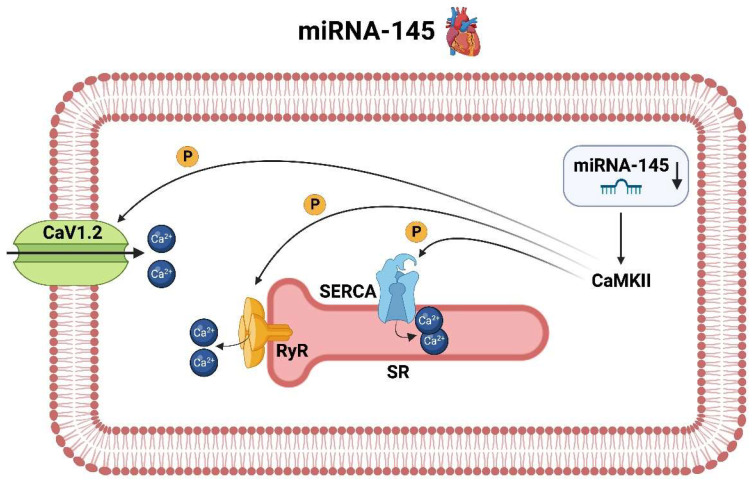
Molecular pathways of miRNA-145 in the context of atrial fibrillation in the heart (created with BioRender.com).

**Figure 9 ijms-25-05568-f009:**
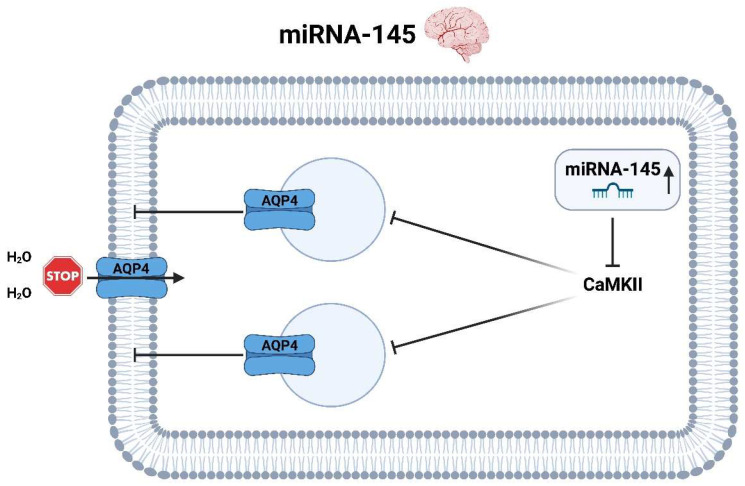
Molecular pathways of miRNA-145 in the context of ischemic stroke in the brain (created with BioRender.com).

**Figure 10 ijms-25-05568-f010:**
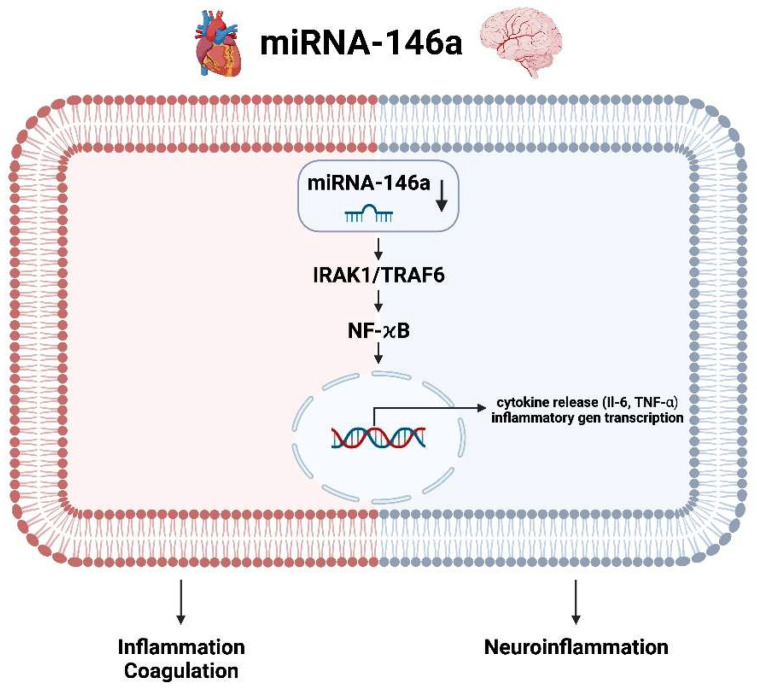
Molecular pathways of miRNA-146a/b in the context of atrial fibrillation in the heart and in the context of ischemic stroke in the brain (created with BioRender.com).

**Table 1 ijms-25-05568-t001:** Overview of the included studies regarding first author, year of publication, study design, relevant miRNAs and major findings.

Authors	Year	Study	miRNA	Major Outcomes
Arroyo et al. [45]	2018	Human	miR-146a	miR-146a Negative regulator of inflammation Downregulation with higher risk of MACE
Benito et al. [46]	2022	Human	miR-1-3p	miR-1-3p Significantly higher plasma levels in patients with cryptogenic stroke and thereby newly diagnosed AF than in patients with sinus rhythm
Chen et al. [47]	2018	Human	miR-15a-5p miR-17-5p miR-19b-3p miR-20a-5p	miR-15a-5p, miR-17-5p, miR-19b-3p, miR-20a-5p Potential biomarkers to differentiate between embolic and thrombotic stroke Association with concomitant diseases of embolic stroke (especially atrial fibrillation)
Kim et al. [48]	2022	Human	miR-93-5p miR-378f miR-450b-5p miR-629-5p	miR-378f, miR-450b-5p Significantly elevated levels in thrombi of patients with cardio-embolic stroke miR-93-5p, miR-629-5p Significantly elevated levels in thrombi of patients with neurological deterioration
Kiyosawa et al. [49]	2020	Human	miR-22-3p miR-128-3p miR-130a-3p miR-140-5p miR-143-3p miR-144-5p miR-148b-3p miR-192-5p miR-194-5p miR-497-5p miR-652-3p	miR-22-3p, miR-128-3p, miR-130a-3p, miR-140-5p, miR-143-3p, miR-497-5p Positive correlation with CHA_2_DS_2_-VASc-Score Upregulation as potential feedback loop against pro-fibrotic mechanisms in patients miR-148b-3p, miR-652-3p Positive correlation with CHA_2_DS_2_-VASc-Score miR-144-5p, miR-192-5p Negative correlation with CHA_2_DS_2_-VASc-Score miR-194-5p Negative correlation with CHA_2_DS_2_-VASc-Score Upregulation as potential feedback loop against pro-fibrotic mechanisms in patients
Koutsis et al. [50]	2013	Review	miR-1 miR-26 miR-328 antag. miR-1 antag. miR-145 antag. miR-181a antag. miR-497	miR-1, miR-26, miR-328 Association with presence of atrial fibrillation Antagonists/Antagomirs of miR-1, miR-145, miR-181a, miR-497 Potential of neuroprotection in in vivo models
Li et al. [51]	2020	Human	miR-1 miR-1-3p miR-21 miR-21-5p miR-155 miR-155-5p miR-192 miR-192-5p	miR-1, miR-1-3p, miR-21, miR-21-5p, miR-155, miR-155-5p miR-192, miR-192-5p, Potential biomarkers with association regarding stroke in AF-patients
Liu et al. [52]	2013	In vivo	miR-21 miR-30c-1 miR-142-3p miR-142-5p miR-146a miR-196a/b/c miR-206 miR-224 miR-290 miR-291a-5p miR-324-3p	miR-21, miR-142-3p, miR-142-5p, miR-146a Significant upregulation during recovery from embolic stroke miR-196a/b/c, miR-224, miR-324-3p Significant downregulation during recovery from embolic stroke miR-206, miR-290, miR-291a-5p, miR-30c-1 Positive correlation with infarct size
Llombart et al. [53]	2013	Review	miR-145 miR-210	miR-145 Modulator of smooth muscle cells Upregulation in stroke miR-210 Downregulation in ischemic stroke
Martinez et al. [54]	2017	Review	miR-124 miR-133b miR-145 miR-200a miR-711 miR-762 miR-1892	miR-200a, miR-145, miR-762, miR-1892 Potential neuroprotective role miR-124, miR-145, miR-711 Predictors to mediate anti-inflammatory and microglia/macrophage activation pathways miR-133b, miR-145 Participants in nerve cell remodeling and functional recovery after stroke
Modak et al. [55]	2019	Human	miR-15b-5p miR-29a-5p miR-145-5p miR-151a-3p miR-487b-3p miR-647 miR-664a-3p miR-943 miR-1273e miR-2116-5p miR-4531 miR-4709-3p miR-4742-3p mirR-4756-3p miR-4764-5p miR-5187-3p miR-5584-3p	miR-145-5p, miR-664a-3p, miR-943 Significant upregulation in ischemic stroke miR-647, miR-664a-3p, miR-1273e, miR-2116-5p, miR-4531, miR-4709-3p, miR-4742-3p, mirR-4756-3p, miR-4764-5p, miR-5187-3p, miR-5584-3p Significant downregulation in ischemic stroke miR-29a-5p, miR-151a-3p, miR-487b-3p Significant upregulation in severe ischemic stroke miR-15b-5p, miR-4531 Significant downregulation in severe ischemic stroke
Rivera-Caravaca et al. [56]	2020	Human	miR-22-3p miR-107 miR-146a-5p	miR-22-3p Upregulation associated with higher risk of MACE miR-107 Upregulation associated with higher risk of MACE miR-146a-5p Downregulation associated with higher risk of MACE
Sadik et al. [57]	2021	Human	miR-155	miR-155 Significantly elevation in patients with ischemic stroke Association between plasma level expression of miR-155 and atrial fibrillation (*p* = 0.040) High diagnostic accuracy for acute, ischemic stroke event (AUC = 0.900; *p* < 0.001)
Tan et al. [58]	2017	Human + In vitro	miR-19a miR-20a miR-185 miR-374b	miR-19a, miR-20a, miR-185, miR-374b Significant upregulation in patients with cardioembolic stroke in comparison to non-cardioembolic stroke Relevant association regarding CD46 mRNA
Wang et al. [46]	2010	Review	miR-1	miR-1 Downregulation with anti-arrhythmic effect and lower risk of cardiovascular events
Wu et al. [25]	2016	Human	miR-18a miR-19b miR-21 miR-25 miR-29b miR-164a miR-186 miR-328 miR-335	miR-164a, miR-335 Regulation of endothelin and coagulation pathways in cardioembolic stroke miR-19b, miR-186, miR-328, miR-335 Regulation of B and T cell activation in cardioembolic stroke Regulation of protein synthesis and turnover in cardioembolic stroke miR-18a, miR-21, miR-25, miR-29b Regulation of protein synthesis and turnover in cardioembolic stroke Regulation of neurodegeneration in cardioembolic stroke Regulation of cell death in cardioembolic stroke Regulation of proliferation in cardioembolic stroke
Xie et al. [59]	2023	Human	miR-30e-5p miR-154-5p miR-641	miR-30e-5p, miR-154-5p, miR-641 Significant upregulation in patients with AF-related stroke in comparison to patients with AF alone
Zheng et al. [60]	2019	Human	miR-21-5p miR-206 miR-491-5p miR-3123	miR-21-5p, miR-206, miR-3123 Significant upregulation in patients with cardioembolic stroke and hemorrhagic transformation in comparison to patients without hemorrhagic transformation miR-491-5p No significant upregulation between patients with cardioembolic stroke and hemorrhagic transformation in comparison to patients without hemorrhagic transformation
Zou et al. [61]	2019	Human	miR-27a-3p miR-27b-3p miR-494-3p	miR-27a-3p: Involved in NF-kappaB signaling pathway Involved in negative regulation myoblast differentiation Potential Biomarker for AF-related stroke detection miR-27b-3p Involved in toll-like receptor signaling pathway Involved in regulation of neuron differentiation Potential Biomarker for AF-related stroke detection miR-494-3p Involved in cardiac muscle cell differentiation Involved in neuron migration and differentiation Potential Biomarker for AF-related stroke detection

## Data Availability

Not applicable.

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
