# Peer review of "Deciphering the Role of microRNAs: Unveiling Clinical Biomarkers and Therapeutic Avenues in Atrial Fibrillation and Associated Stroke—A Systematic Review"

_ijms, 2024, doi:10.3390/ijms25105568_

Round 1

Reviewer 1 Report

Comments and Suggestions for Authors

The manuscript proposed by Boxhammer and co-workers, entitled „Deciphering the Role of microRNAs: Unveiling Clinical Biomarkers and Therapeutic Avenues in Atrial Fibrillation and Associated Stroke – A systematic review” (ijms-2993853) is a review. The objective of this paper was to provide a compilation of previously established miRNAs to demonstrate which miRNAs can be employed in the clinical setting for the early detection of a cardioembolic event, and which may also possess significant therapeutic potential after stroke due to atrial fibrillation in the future.

The manuscript is interesting, but it needs to be modified before it can be published.

Kindly find below my principal remarks. 

- the presented evidence for the innovative nature of the work is not fully sufficient, similar literature studies, even more extensive ones, are already available

- the presented analysis of the role of individual miRNAs is based on literature data, which does not consider the latest scientific studies or previous works in which information on this subject can be found

- the authors do not cite relevant scientific works on the subject that are important to the manuscript

- the technical organization of work should be better, captions under the drawings appear on the side along with the text, which makes analysis difficult

- the work lacks a presentation of the directions in which such research could develop, it is necessary to define possible scenarios, provide possibilities and limit the importance of the topic

Comments on the Quality of English Language

Minor editing of English editing is required

Reviewer 2 Report

Comments and Suggestions for Authors

The purpose of the present review was to summarize 18 existing manuscripts that have dealt with this combined topic of atrial fibrillation (AF)  and associated acute ischemic stroke (AIS) in detail and to shed light on the most frequently mentioned miRNAs -1, -19, -21, -145, and -146 with regard to their molecular mechanisms and targets on both the heart and the brain. As a results, authors suggested that possible diagnostic and therapeutic consequences for the future could be derived. This manuscript is written well, however this will be able to revise jus a few points. 

1) Regarding Figure 1, could you add the abbreviation for these? In addition, it may not be able to easy to understand. 

2) Regarding Table 1, could you explain more exactly? What is this main outcome in these references? 

3) With regard to Figures, there is not clear all of them, so could you revise these figures more clearly?  There are many figures in this manuscript, could you join there and more clearly for readers? 

4) With regard to references, could you confirm and revise these references by Journal guidelines?

Comments on the Quality of English Language

N/A

Round 2

Reviewer 1 Report

Comments and Suggestions for Authors

The revised version of the manuscript proposed by Boxhammer and co-workers meets my requirements. The authors presented valuable comments to all my questions and doubts. Therefore, in my opinion, the manuscript may be published in IJMS.